# Thermal and Optical Properties of Porous Nanomesh Structures for Sensitive Terahertz Bolometric Detection

**DOI:** 10.3390/s22145109

**Published:** 2022-07-07

**Authors:** Ryoko Yamamoto, Akira Kojima, Nobuyoshi Koshida, Isao Morohashi, Kazuhiko Hirakawa, Ya Zhang

**Affiliations:** 1Institute of Engineering, Tokyo University of Agriculture and Technology, 2-24-16 Naka-cho, Koganei-shi 184-8588, Japan; s217738u@st.go.tuat.ac.jp (R.Y.); kojima_ak@quantum14.com (A.K.); koshida@cc.tuat.ac.jp (N.K.); 2National Institute of Information and Communications Technology, 4-2-1 Nukui-Kitamachi, Koganei-shi 184-8795, Japan; morohashi@nict.go.jp; 3Institute of Industrial Science, University of Tokyo, 4-6-1 Komaba, Meguro-ku 153-8505, Japan; hirakawa@iis.u-tokyo.ac.jp; 4Institute for Nano Quantum Information Electronics, University of Tokyo, 4-6-1 Komaba, Meguro-ku 153-8505, Japan

**Keywords:** porous nanostructure, thermal conductance, terahertz sensing, terahertz absorptance, MEMS resonator, bolometer

## Abstract

Terahertz (THz) electromagnetic waves are attractive for use in nondestructive and biocompatible sensing applications. Thermal sensors are widely used for THz detection owing to the small photon energies of THz radiation, where this requires materials with low thermal conductivity and a small heat capacity to ensure the sensitive and fast operation of the sensors. In this study, we investigated the thermal and optical properties of porous nanomesh structures for sensitive THz bolometric detection. Nanometer (nm)-scale hole array structures were formed on gallium arsenide (GaAs) microelectromechanical system (MEMS) beams to improve their thermal properties. The thermal conductance of the porous MEMS beams was obtained by measuring their thermal bandwidths; it was found to decrease by as much as ~90% when the porosity (*P*) of the porous nanostructure was increased to ~0.69. We also measured the THz absorptance of the porous hole array structure. The results show that although the porous nanostructure has a much smaller area than the bulk material, it maintained a high coefficient of THz absorptance because the featured size was much smaller than the THz wavelength. The measured absorptance agreed well with that calculated by using the Drude model. These results demonstrate that the porous nanomesh structure is promising for developing highly sensitive THz thermal sensors.

## 1. Introduction

Terahertz (THz) electromagnetic waves are attractive for use in spectroscopy and imaging applications owing to the rich spectral features in the THz frequency band [1]. Uncooled, sensitive, and fast THz sensors are crucial for compact THz spectroscopy [2,3,4,5] and imaging systems [6,7,8]. However, owing to the small photon energies involved, THz photodetectors that use intersublevel transitions in low-dimensional nanostructures [9,10] or impurities [11] in semiconductors cannot operate at room temperature. The thermal sensor [12,13] is another type of THz sensor that converts the input light into heat and detects the induced rise in temperature by using a thermistor to determine the input light-induced power. Commonly used thermal sensors include pyroelectric sensors, vanadium oxide (VOx) bolometers [14,15,16], and MEMS bolometers [17,18,19,20]. They have been proposed for THz and infrared detection at room temperature.

The authors of this study developed an uncooled, sensitive, and fast THz bolometer in past work by using a GaAs-based double-clamped MEMS beam resonator [18,19]. The absorption of an incident THz electromagnetic wave induces thermal stress in the MEMS beam, which reduces the resonance frequency of the MEMS resonator. The MEMS resonator detects such a shift in the resonance frequency and works as a very sensitive bolometer. It was found that the MEMS resonator has an ultra-high temperature sensitivity of ~1 μk [19]. This renders the MEMS resonator among the most sensitive uncooled thermistors, to the best of our knowledge. Furthermore, to achieve a fast THz detection, the MEMS resonator is operated in a self-oscillation mode by using a phase-locked loop (PLL). The PLL tracks the resonance frequency of the MEMS resonator and outputs a voltage signal that is proportional to the shift in the resonance frequency. The detection speed is up to ~10 kHz [19], which is limited by the demodulation bandwidth of the PLL and the thermal decay process in the MEMS beam.

In general, it is difficult for thermal sensors to simultaneously achieve high sensitivity and fast operation. High sensitivity requires that the structure of the bolometer have a low thermal conductance (*G*_T_) to increase the rise in temperature induced by irradiation by light. However, a low thermal conductance increases the duration of thermal decay *τ*_D_ = *C*_T_/*G*_T_, where *C*_T_ expresses the heat capacity of the bolometer; thus, this reduces the speed of operation of the bolometer [19].

A nanometer (nm)-sized hole array structure fabricated on a MEMS beam can suppress thermal conductance and reduce heat capacity [21,22,23,24]. Such a structure reduces the total volume of the MEMS beam to reduce both its thermal conductance and heat capacity. Furthermore, when the featured size of the porous nanostructure is smaller than the mean paths of phonons, this reduces the thermal conductance of the MEMS beam, as well as the thermal conductivity [22,23] of the material, to improve its thermal sensitivity. Previously, we fabricated nanohole array structures with a porosity (*P*) of 0.2–0.3 on MEMS beams to improve their thermal sensitivities [24]. High-porosity nanohole array structures that could more significantly suppress the thermal conductance of the MEMS beams were not achieved owing to the fabrication difficulties. Moreover, the porous structures of high porosities may lead to drawbacks for MEMS bolometers that have not been evaluated. Because the porous nanohole array structure reduces the volume of the MEMS beam, it may reduce its mechanical strength to degrade the quality (Q-) factor of the MEMS resonator. Furthermore, it may reduce the surface area that can be used to absorb light, where this is detrimental to the performance of the light sensor. Both the positive and negative effects of the use of porous nanohole array structures in THz/infrared sensing applications thus need to be investigated.

In this study, we investigated the thermal and optical properties of porous nanomesh structures of high porosities for sensitive THz bolometric detection. Porous nm-scale hole array structures were formed on GaAs MEMS beams to improve their thermal properties. The thermal conductance of the porous MEMS beams was obtained by measuring their thermal time constant. We found that the thermal conductance of the MEMS beam decreased significantly, by as much as ~90%, when *P* of the porous nanostructure was increased to ~0.69. Furthermore, we measured the THz absorptance of the porous hole array structure. The results show that although the porous nanostructure had a much smaller volume than the bulk material, it could maintain a high coefficient of THz absorptance because the featured size was much smaller than the THz wavelength. The measured absorptance agreed well with that calculated by using the Drude model. These results demonstrate that the porous nanostructure is promising for developing highly sensitive THz thermal sensors.

## 2. Mechanical and Thermal Properties of Porous Nanostructures

We determined how the thermal conductance and mechanical strength of the MEMS beam were affected by the porous nanostructure by using the finite element method (COMSOL Multiphysics 5). Because we discuss a periodic structure here, we chose only a single component from the hole array for the calculation, as schematically shown in Figure 1a. We considered two designs for the porous nanostructure, with a round hole and a square hole as the basic units. We assumed that the basic unit shown in Figure 1a had a size of 1 μm × 1 μm × 1 μm, and the size of the hole ranged from 0 to 1 μm. Thus, the passivity of the structure, *P*, changed from 0 to ~0.79 for the round hole structure and from 0 to 1 for the square hole structure. To calculate changes in the thermal conductance of the structure, we assumed that there was a temperature difference of Δ*T* = 1 K between the two edge surfaces of the basic unit. We then calculated the heat flux *H* so that the thermal conductance of the basic unit between the edges was derived as *G*_T_ = *H*/Δ*T*. Figure 1b,c show the calculated distribution of heat flux for units with round and square holes, respectively. The derived normalized thermal conductance is plotted in Figure 1f, where the solid and dashed curves represent the thermal conductance of the round hole design and the square hole design, respectively. The thermal conductance decreased quickly with the increase in *P*. For the round hole structure, the thermal conductance dropped to zero when *P* = π/4 ≈ 0.79, where the diameter of the hole was equal to the width of the unit, and the structure will be cracked in reality. On the contrary, the square hole structure yielded a higher *P*, approaching ~1.

We also calculated changes in the mechanical strength of the structure. By assuming that it was stretched by a small force *F*, we calculated only the tensile strength of the structure based on the deformation σ and *F*, as *E* = *F*/σ. The calculated distributions of stress in the round and square hole structures are shown in Figure 1d,e, respectively. Both structures exhibited similar reductions in tensile strength when *P* was less than 0.5, as shown in Figure 1g. However, when *P* approached π/4, the tensile strength dropped to zero for the round hole structure, which was similar to the behavior of its thermal conductance (see Figure 1f). Because we wanted to obtain as high a value of *P* as possible to simultaneously reduce the heat capacitance and the thermal conductance, we fabricated samples by using the square hole structure, i.e., the nanomesh structure.

## 3. Fabrication of MEMS Beams with Porous Nanomesh Structures

We grew a modulated AlGaAs/GaAs heterostructure on a GaAs substrate through molecular beam epitaxy [25] to fabricate MEMS beams with a porous nanomesh structure. The wafer structure is schematically shown in Figure 2a. After growing a 200-nm-thick GaAs buffer layer and a 3-μm-thick Al_0.7_Ga_0.3_As sacrificial layer on a semi-insulating (100) GaAs substrate, we formed the beam layer by depositing a 100-nm-thick GaAs layer, a 20-period GaAs/Al_0.3_Ga_0.7_As superlattice structure, and a 400-nm-thick GaAs layer. A mesa layer with a 20-nm-thick GaAs layer, a 55-nm-thick Si-doped Al_0.3_Ga_0.7_As layer, and a 10-nm-thick GaAs capping layer were formed on the beam layer. In past work, we used the mesa layer to form piezoelectric capacitors to electrically drive and detect the oscillations of the MEMS beam. However, in this research, we did not deposit any electrode on the MEMS beam to avoid the influence of metal electrodes on measurements of its thermal conductance. Instead of piezoelectric capacitors to drive and detect oscillations in the MEMS beam, we placed the beam on a piezoelectric block to excite vibrations and probed the induced oscillations by using a laser Doppler vibrometer.

Figure 2b shows the process of fabricating the MEMS beams using porous nanomesh structures. We first formed the nanomesh patterns by using electron beam (EB) lithography. The design of the patterns is shown in the inset of Figure 2b. The size of each square hole was set to 1 µm × 1 µm, and the linewidth of the mesh was modulated from 200 nm to 500 nm, yielding *P* ranging from ~0.44 to ~0.69. Following this, the holes of the nanomesh were etched by a dry-etching facility (SAMCO RIE-101iPHJ) with process gas flows of 2 sccm of Cl_2_, 2 sccm of SiCl_4_, and 10 sccm of Ar. We applied an RF ICP power of 150 W with a bias power of 30 W for 80 s to etch the nanomesh structure. The MEMS beam was then shaped by using photolithography and wet etching, and the sacrificial layer (Al_0.7_Ga_0.3_As) was selectively removed by using diluted hydrofluoric acid [25]. To prevent the highly porous MEMS beams from cracking, we used a supercritical drying process to release them. Figure 2c shows a microscope image of the fabricated MEMS beam with a porous nanomesh structure (*P* = ~0.69), and the inset shows a blown-up image taken by a scanning electron microscope (SEM). It shows details of the nanomesh structure. All the MEMS beams had the same geometry (length: ~100 μm; width: ~30 μm; thickness: 0.6 μm). As seen, the nanomesh was homogenously formed on the MEMS beam. Figure 2d shows the surface profile of the MEMS beam that was measured by using a confocal laser microscope (Olympus OLS4100). The fabricated porous beam was bent slightly downward, and the center of the sample had a deflection of ~400 nm. This is because the porous MEMS beam had a lower mechanical strength than the bulk MEMS beam without a hole array; thus, it was more easily affected by its internal residual stress [26,27,28], which might have originated from the slightly different lattice constants of GaAs and aluminium arsenide (AlAs).

## 4. Results

### 4.1. Mechanical Resonance of Porous MEMS Beams

Figure 3a shows the experimental setup used to measure the mechanical oscillations and thermal conductance of the MEMS beam. The entire MEMS chip was placed on a piezoelectric block that was used to drive vibrations in it. The induced vibrational motion was monitored with a laser Doppler vibrometer. The oscillation spectra were measured by a lock-in amplifier with a built-in PLL (Zurich HF2LI) [19]. To measure the thermal response of the MEMS resonator, we used a second laser with a wavelength of ~530 nm to heat the MEMS beam. Its output power could be modulated with an input transistor–transistor logic (TTL) signal. The output power of the laser was ~5 mW, and we used a light attenuator to reduce the power to serval μW so as to avoid the overheating of the MEMS beam. We carefully aligned the laser heating spot with the geometric center of the MEMS beam. All the measurements were performed at room temperature in a vacuum of ~10^−5^ torr.

Figure 3b shows the measured resonance spectrum of a reference MEMS beam without a nanomesh structure. The resonance frequency was ~119 kHz and the quality (Q) factor was ~1000. Figure 3c–e show the measured resonance spectra of three porous MEMS beams with nanomesh structures with *P* = 0.44, 0.56, and 0.64, respectively. The porous MEMS beams had a resonance frequency of 176.5 kHz, 165 kHz, and 175 kHz, and Q-factors of ~900, ~1000, and ~500, respectively; this indicates that the highly porous nanomesh structures did not significantly degrade the mechanical quality of the beam.

### 4.2. Duration of Thermal Decay

The MEMS beam resonators were driven in self-oscillation mode by using the PLL for fast frequency modulation, as reported in our previous works [18,24]. When the 530 nm laser was switched on, light was absorbed by the MEMS beam and it was heated owing to the photothermal effect, which reduced its resonance frequency. Figure 4a shows the shift in resonance frequency (Δ*f)* when the laser was periodically modulated. Δ*f* was measured as a function of the modulation frequency of the laser by using a lock-in amplifier, as shown in Figure 4b. The solid dots and the plus, square, and triangle signs represent the normalized values of Δ*f* of the reference beam, as well as beams with *P* values of 0.44, 0.64, and 0.69, respectively. It is notable that the effect of the PLL was eliminated by normalizing all the data with the frequency response curve of the PLL. The reference MEMS beam without holes had the highest bandwidth (−3 dB: ~2 kHz), and the bandwidth decreased with an increase in *P*.

We considered a simple model of thermal decay. The shift in frequency can be expressed as
(1)$$\begin{array}{c} \Delta f=\frac{\Delta f_{o}}{\sqrt{(1+{\left(2\tau_{D}\pi f_{m}\;\right)}^{2}\;}} \end{array}\;\;\;\;\;\;\;\;\;$$
where *τ*_D_ and *f*_m_ indicate the duration of thermal decay of the MEMS beam and the modulation frequency of light, respectively. Δ*f*_0_ indicates the frequency response to the static heat input. This model is reasonable because the laser spot was very small (diameter: ~5 μm) compared with the size of the MEMS beam (length: ~100 μm; width: ~30 μm). The solid lines in Figure 4b represent the curves of fitting of the model with the experimental data by using Equation (1). All the data fitted well with the simple model of thermal decay. *τ*_D_ were ~76.1 μs, ~165 μs, ~198 μs, and ~318 μs for the reference beam, and beams with *P* values of 0.44, 0.64, and 0.69, respectively.

The thermal decay time can be expressed as
(2)$$\tau_{D}=\frac{C_{T}\left(p\right)}{G_{T}\left(p\right)}=\frac{PC_{T}\left(0\right)}{G_{T}\left(p\right)}$$
where *p* indicates the *P* of the MEMS beam and *C*_T_(0) indicates the heat capacity of the reference MEMS beam. The thermal conductance *G*_T_(*p*) of the porous beam, normalized by the thermal conductance of the reference beam *G*_T_(0), is then given by
(3)$$\frac{G_{T}\left(p\right)}{G_{T}\left(0\right)}=\frac{C_{T}\left(p\right)}{G_{T}\left(p\right)}=\frac{P\tau_{D}\left(0\right)}{\tau_{D}}$$
where $\tau_{D}\left(0\right)$ indicates the thermal decay time of the reference MEMS beam. 

The red square dots in Figure 4c are used to plot the normalized thermal conductance of the MEMS beams with a porous nanomesh. For comparison, the thermal conductances of the round hole array at small *P* are plotted using the solid triangular dots in Figure 4c. They were reported in a previous publication [24]. The thermal conductance decreased significantly as *P* increased. When the *P* was increased to ~0.69, the thermal conductance of the porous nanomesh beam decreased by over 90% compared with that of the reference beam; this was a much larger reduction than reported in our previous work (<50%) at small *P*. Because thermal sensitivity is inversely proportional to thermal conductance, an enhancement in it of over 10 times was expected for highly porous MEMS beams. The solid and dashed curves in Figure 4c show the thermal conductance of the nanomesh and the round hole array structures, respectively, calculated by using the finite element method as described in Section 2. The measured thermal conductance agreed well with the numerical value, indicating that the geometry of the structure was still the main factor influencing the reduction in thermal conductance. On the contrary, for the MEMS beam with *P* = 0.69, the experimental thermal conductance notably differed from the numerical result (solid curve in Figure 4c), suggesting that the phononic effect might have been active in highly porous MEMS beams. This requires further investigation in future work. 

## 5. THz and Infrared Absorption of the Nanomesh Structure

Although the porous nanomesh structure has adequate mechanical and thermal properties, as shown in Section 2 and Section 3, there is concern regarding its coefficient of optical absorptance. This is because the removal of materials to form a nanomesh structure significantly reduces the surface area for light absorptance, as schematically shown in Figure 5a. However, because the featured size of the nanomesh, i.e., the size of the hole (~1 μm), is much smaller than the wavelengths of THz electromagnetic waves (300 μm for 1 THz), we expect the nanomesh structure to have the same, high THz absorptance using a simple metal film [29].

First, we theoretically calculated the THz/infrared absorptance coefficient for a simple metal film. Thin metal films have long been used as THz/infrared absorbers [30]. A THz/infrared absorptance coefficient of 50% can be achieved for a self-supported metal film when its sheet resistivity matches the vacuum impedance of electromagnetic waves. However, for a thin metal film with a supporting substrate, the absorptance decreases owing to the increased reflection by the substrate. Such absorptance is adequately described by the Drude model by considering the interference of multiple reflected light [30]. We assumed that the plasma frequency of the film material was much higher than the frequency of the input THz/infrared light, which is applicable to most types of metals. The refraction index of the supporting substrate was *n* = 3.6. The transmission was taken just below the metal film so that the absorptance in the substrate was not considered. The calculated transmittance, reflectance, and absorptance are plotted as a function of sheet resistivity in Figure 5b. The absorptance reached a peak of ~23% when the sheet resistivity was ~90 Ω, which agrees well with our previous experimental results [19].

We then fabricated a sample metal nanomesh and a sample metal film by depositing a 15-nm-thick film of nickel by using electron beam evaporation on a high-resistivity silicon substrate. The same substrate was used as a reference sample, as schematically shown in Figure 6a. Figure 6b shows an SEM image of the fabricated nanomesh sample. The size of the square hole in the mesh was 5 μm and the linewidth of the mesh was 500 nm. We chose this design to explore the coefficient of light absorptance when the incident wavelength was much larger than, close to, or smaller than the size of the hole of the mesh. We used high-resistivity silicon rather than GaAs because the latter involves complicated light–phonon interactions that may affect the THz/infrared absorptance. This makes it difficult to study absorptance based on the mesh structure.

The optical measurement was performed with a Fourier Transform infrared spectrometer (Jasco FT/IR 6100). Figure 6c shows the measured spectra of the sample of the Ni film in the far-infrared range (60–600 cm^−1^), and the measured transmission (*T*) and reflection (*R*) spectra are shown by the black and blue curves, respectively. The red curve in Figure 6c shows the absorptance spectra *A* = 1 − *T* − *R*. The absorptance coefficient was ~10% and remained almost constant in the far-infrared range. We calculated the sheet resistance as ~10 Ω from the absorptance coefficient (see Figure 5b); this agreed well with the sheet resistance calculated based on the resistivity of Ni and the thickness of the film. The slight increase in absorptance in the high-frequency band (>400 cm^−1^) occurred because of the absorptance in the substrate.

The equivalent electrical conductance of the nanomesh structure decreases with increasing *P*, and generally follows the same rule as the thermal conductance in Figure 1f. This is because the changes in both occurred due to a reduction in the cross-section of the mesh structure. *P =* (5 μm)^2^/(5.5 μm)^2^ ≈ 0.83 for the given design parameters (hole width: 5 μm; linewidth: 500 nm). The equivalent electrical conductance of the mesh structure decreased to ~10% of that of the film, yielding a sheet resistance of ~100 Ω. This was very close to the optimized sheet resistance that induces the peak THz/infrared absorptance (see Figure 5b). Figure 6d plots the measured far-infrared absorptance values of the high-resistivity Si substrate, the Ni film sample, and the nanomesh sample, shown as the black, blue, and red curves, respectively. At a low frequency band (<200 cm^−1^), the nanomesh maintained a high absorptance coefficient of over 20%, much higher than those of the substrate (nearly zero) and the Ni film (~10%). This absorptance coefficient was close to the maximum value of the calculated absorptance coefficient for a metal film with a supporting Si substrate (see Figure 5b). The result indicates that although the nanomesh structure might have a much smaller area than the bulk metal film (in this case, 83% smaller), THz absorptance was still as high as that of an optimized metal film absorber as long as the equivalent sheet resistance achieved the optimized value.

In addition to the equivalent sheet resistance, the relation between the featured size of the nanomesh and the incident wavelength was an important parameter to achieve high absorptance. To calculate the “equivalent sheet resistance” of the nanomesh structure, we assumed that the wavelength was much larger than the featured size of the nanomesh, i.e., the size of the hole, and the width of the Ni line. Then, the THz/infrared wave could not pass through the holes of the nanomesh. However, as the frequency of incident light increased, the wavelength decreased to the same order as the size of the hole. Then, the incident THz/infrared wave could pass through the hole arrays to reduce absorptance by the nanomesh structure. As shown in Figure 6d, the measured light absorptance of the nanomesh structure gradually decreased when the wavelength of incident light exceeded 200 cm^−1^ and reached its minimum value at ~450 cm^−1^. This is reasonable because at 450 cm^−1^, the wavelength of light in the Si substrate decreased to 22.2 μm/3.6 ≈ 6 μm, which was close to the size of the hole of the fabricated nanomesh structure.

The measured absorptance increased again above 450 cm^−1^ owing to the increased absorptance in the substrate that originated from light–phonon interactions in the Si material, as shown by the black curve in Figure 6d. Absorption by the substrate achieved a peak at ~600 cm^−1^. The absorption of light in the substrate is usually not desirable for bolometric applications, particularly when a solid immersed silicon lens is used, and the light is incident from the substrate side of the bolometer device. The absorption of the substrate reduces the light that could arrive at the metal absorber.

At even higher frequencies in the middle-infrared band, the absorption of light by the nanomesh gradually became a combination of absorption by the substrate (*A*_sub_) and the metal (*A*_metal_) because the size of the hole of the fabricated nanomesh structure was larger than the wavelength of light:*A*_mesh_ = *A*_sub_ × *P* + *A*_metal_ × (1 − *P*)(4)

Figure 6e plots the absorptions of light by the substrate, Ni film, and nanomesh samples in the middle-infrared band (1000–7000 cm^−1^), represented by the black, blue, and red solid curves, respectively. The black dashed curve in Figure 6e plots the combined absorption, obtained by using Equation (4). The measured infrared absorption of the nanomesh structure gradually converged to the calculated *A*_mesh_ at ~5000 cm^−1^. This is because at 5500 cm^−1^, the wavelength of light in the Si substrate decreased to 500 nm, which was identical to the width of the metal lines in the fabricated nanomesh structure.

These results indicate that as long as the wavelength of the incident light is larger than the size of the holes of the porous nanomesh structure, the metal mesh can be regarded as a film absorber and its high *P* does degrade the coefficient of optical absorption. For the nanomesh structure detailed in Section 2 and Section 3, the size of the holes is 1 μm. The deposition of a metal film with appropriate thickness on the porous nanomesh structure yields a good THz or middle-infrared absorber. This can be extended to the near-infrared or visible-light bands if we further reduce the size of the hole of the nanomesh in the fabrication.

## 6. Conclusions

In this study, we investigated the thermal and optical properties of porous nanomesh structures for the sensitive bolometric detection of THz waves. Nanometer-scale hole array structures were formed on GaAs MEMS beams to improve their thermal properties. The thermal conductance of the porous MEMS beams was obtained by measuring their thermal time constant. The thermal conductance of the MEMS beam was significantly reduced, by as much as ~90%, when the *P* of the mesh was increased to ~0.69. Furthermore, we measured the THz/infrared absorptance of the porous hole array structure. The results showed that although the porous nanostructure had a much smaller volume than the bulk material, it maintained a high coefficient of THz absorptance because the featured size was much smaller than the THz wavelength. The measured absorptance agreed well with that calculated by using the Drude model. These results demonstrate that the porous nanostructure is promising for realizing highly sensitive THz thermal sensors.

## Figures and Tables

**Figure 1 sensors-22-05109-f001:**
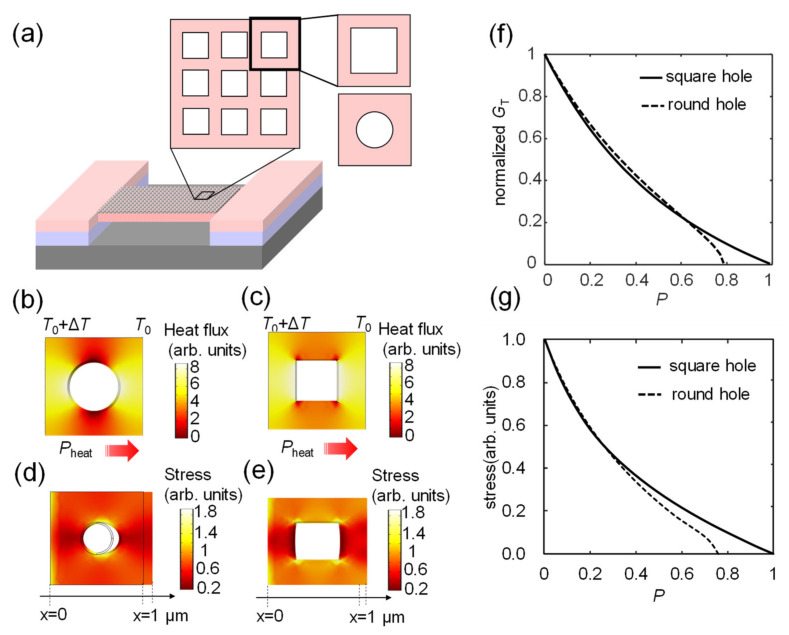
(**a**) The basic unit of a porous MEMS resonator. The holes were designed by using a round hole array and a nanomesh structure. (**b**) The calculated distribution of heat flow of the round hole structure, obtained by assuming a fixed temperature difference Δ*T* = 1 K between the left and right edges of the round hole unit. (**c**) The calculated distribution of heat flow of the square hole structure. (**d**) The calculated stress distribution of the round hole structure, obtained by assuming a fixed stress in the lateral direction of the unit. (**e**) The calculated stress distribution of the square hole structure. (**f**) The calculated thermal conductance as a function of *P* of the nanostructures. The solid line represents the thermal conductance of the round hole structure, and the dotted line represents that of the square hole structure. (**g**) The calculated tensile strength as a function of *P* of the structure. The solid and the dotted lines represent the tensile strengths of the square and round hole structures, respectively.

**Figure 2 sensors-22-05109-f002:**
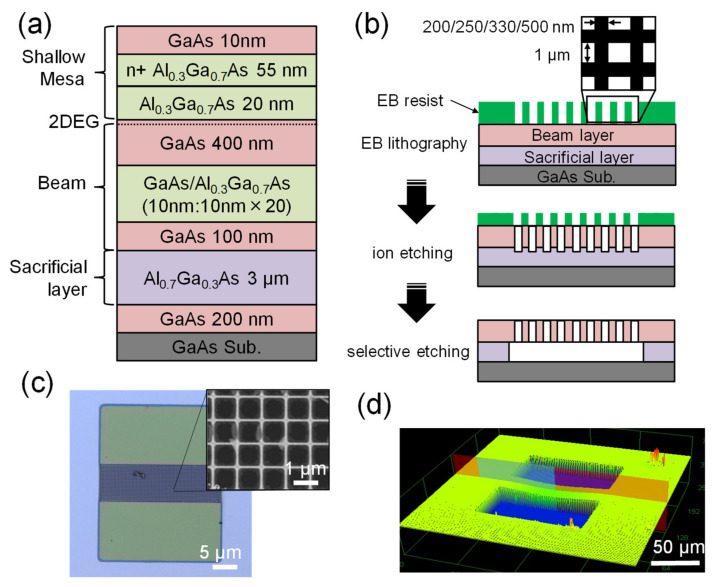
(**a**) The schematic wafer structure for the fabrication of the porous MEMS beams. (**b**) Fabrication of the MEMS beams with porous nanomesh structures. The size of the square hole was fixed at 1 µm × 1 µm, and the linewidth of the mesh was modulated from 200 nm to 500 nm, yielding a range of *P* from ~0.44 to ~0.69. (**c**) A microscope image of a fabricated MEMS beam with a porous nanomesh structure (*P* = 0.69). The inset shows a blown-up image taken by a scanning electron microscope to demonstrate details of the nanomesh structure. (**d**) The surface profile of the MEMS beam measured with a confocal laser microscope. The center of the beam is bent downward by ~400 nm.

**Figure 3 sensors-22-05109-f003:**
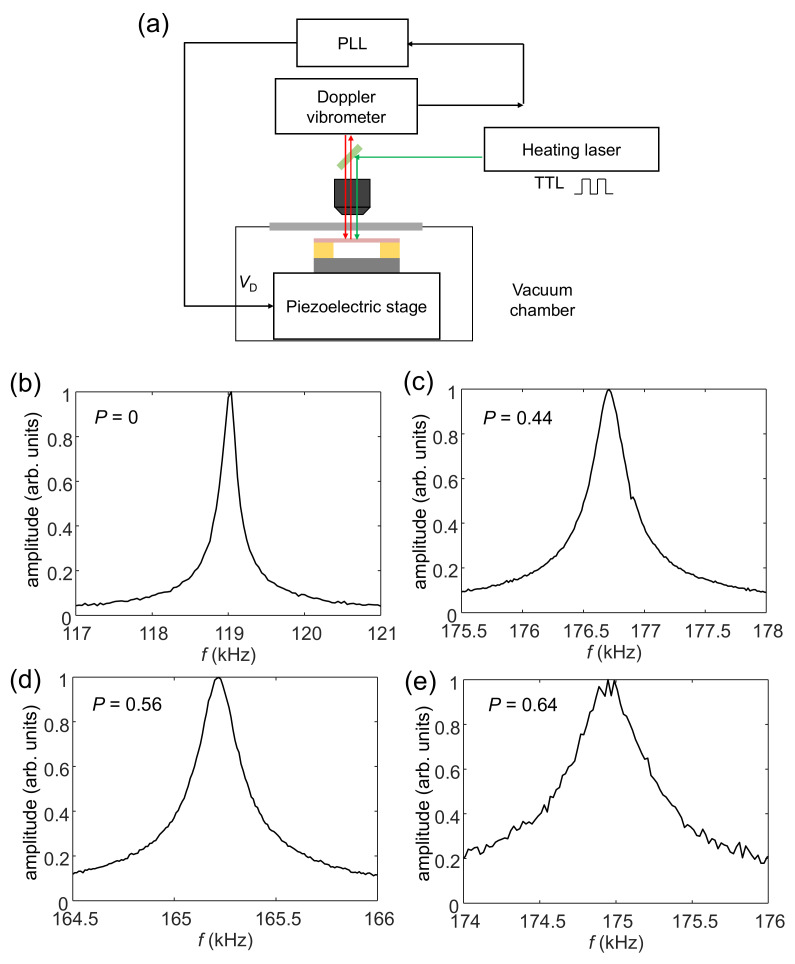
(**a**) The experimental setup used to measure the mechanical oscillations and thermal conductance of the MEMS beam. A piezoelectric block was used to drive vibrations in the MEMS beam, and the induced oscillatory motion was monitored with a laser Doppler vibrometer. A lock-in amplifier with a built-in PLL was used to measure the oscillation spectra. A 520 nm TTL-modulated laser diode was used to heat the MEMS beam. (**b**) The resonance spectrum of a MEMS beam without any nanomesh structure for the first bending mode, of which the resonance frequency was ~119 kHz and the quality(Q)-factor was ~1000. (**c**) The resonance spectra with nanomesh structures (*P* = 0.44) at a resonance frequency of ~176.5 kHz and a Q-factor of ~900. (**d**) The resonance spectrum with a nanomesh structure (*P* = 0.56) at a resonance frequency of ~165 kHz and a Q-factor of ~1000. (**e**) The resonance spectrum of the nanomesh structure (*P* = 0.64) at a resonance frequency of ~175 kHz and a Q-factor of ~500.

**Figure 4 sensors-22-05109-f004:**
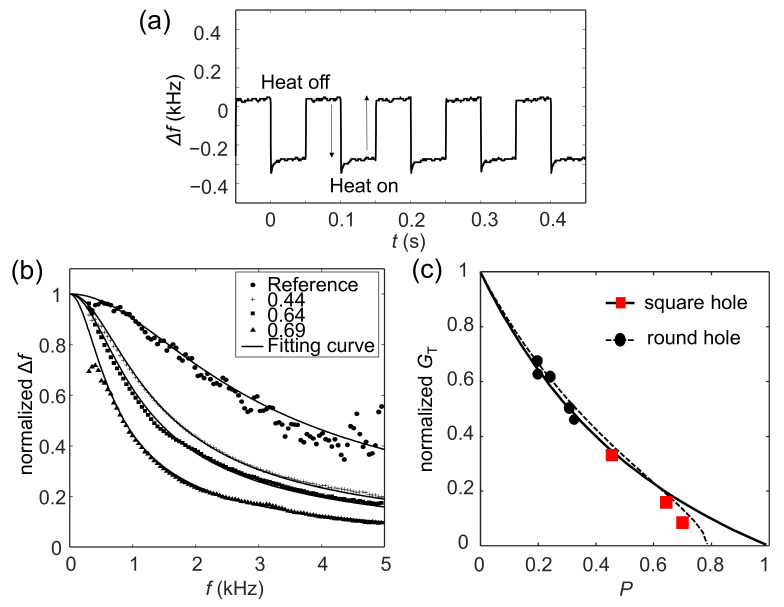
(**a**) The shift in the resonance frequency (Δ*f*) when the laser was periodically modulated. (**b**) Δ*f* measured by using a lock-in amplifier as a function of the modulation frequency of the laser. The solid dots, plus signs, square signs, and triangle signs indicate the normalized Δ*f* for the reference beam, and nanomesh beams with *P* = 0.44, 0.64, and 0.69, respectively. The solid line shows the curves of fitting with the experimental data. (**c**) The red squares show the normalized thermal conductance of MEMS beams with the porous nanomesh. The black dots show the thermal conductance at small *P* with the round hole array structure. The theoretical values calculated with the finite element methods are plotted as the solid and dashed curves for the nanomesh (square hole) and the round hole designs, respectively.

**Figure 5 sensors-22-05109-f005:**
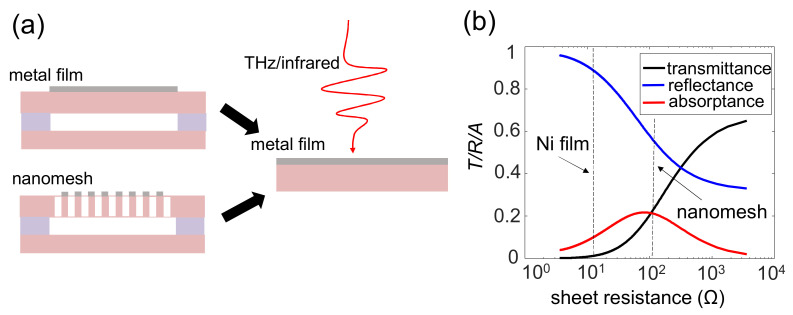
(**a**) The schematic structure applied to a metal film as a light absorber in buck MEMS and MEMS bolometers with a porous nanomesh. When the wavelength of light (THz/infrared) was much larger than the featured size of the nanomesh, the absorptance in both structures was treated as a simple metal film deposited on a supporting substrate. (**b**) The calculated transmission, reflection, and absorptance as a function of sheet resistivity for a simple metal film deposited on a substrate with a refraction index of *n* = 3.6.

**Figure 6 sensors-22-05109-f006:**
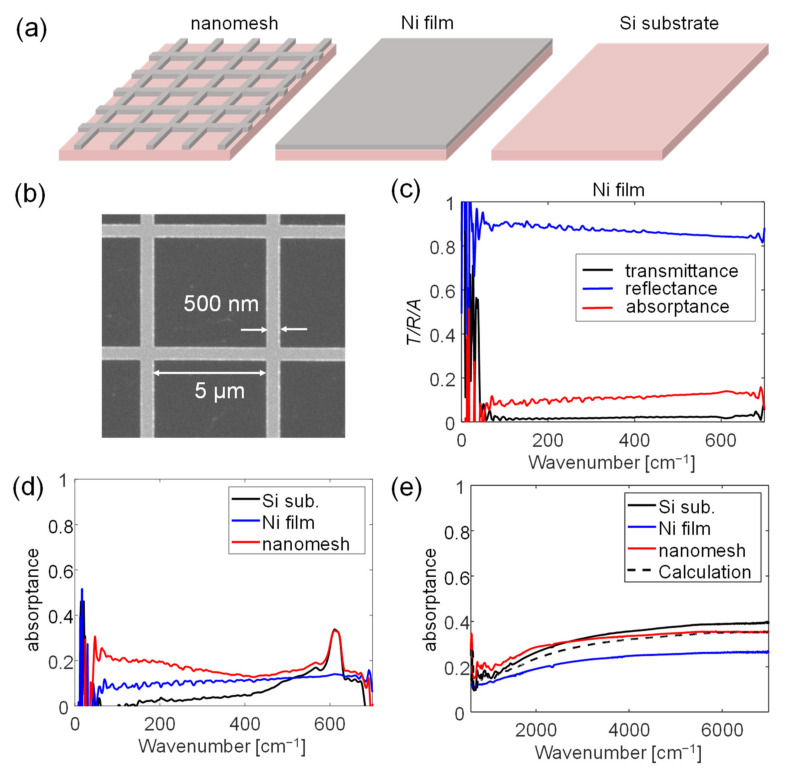
(**a**) Schematic diagrams of a nanomesh absorber, an Ni metal film, and a high-resistivity Si substrate. (**b**) An SEM image of the fabricated nanomesh sample with square holes in the mesh. The size of the square holes was 5 µm × 5 µm, the linewidth of the mesh was 500 nm, and the thickness of the Ni film was ~15 nm. (**c**) The measured spectra of the Ni film in the far-infrared range (60–600 cm^−1^). The measured transmission (*T*) and reflection (*R*) spectra are shown by the black and blue curves, respectively, and the red curve plots the absorptance *A* as calculated by *A* = 1 − *T* − *R*. (**d**) The measured far-infrared absorptance of the highly resistive substrate, the Ni film, and the nanomesh sample are plotted as the black, blue, and red curves, respectively. (**e**) The light absorptance of the substrate, Ni film, and nanomesh samples in the middle-infrared band (1000–7000 cm^−1^), shown by the black, blue, and red solid curves, respectively. The black dashed curve shows the combined absorptance calculated by using Equation (4).

## Data Availability

The data presented in this study are available on request from the corresponding author.

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
