# Peer review of "Thermal and Optical Properties of Porous Nanomesh Structures for Sensitive Terahertz Bolometric Detection"

_sensors, 2022, doi:10.3390/s22145109_

Round 1
Reviewer 1 Report
The manuscript is clearly, but I have some questions:
1. Why is the heat flux distribution of round hole in Fig. 1 different from in your previous research (ref. 24)?
2. Why is the square hole stronger than round hole? Do you can detail explain this problem due to normally compression and stress of round hole stronger than square hole?
3. How to measure the porosity (P) value of hole round and square hole?
4. In Fig.4a, the value of frequency (f) and time (t) is negative.
Reviewer 2 Report
In this manuscript, the authors successfully presented an analysis of thermal properties for two-dimensional nanometer-scale hole array structures on silicon substrate for future utilization in gallium arsenide microelectromechanical system/MEMS beam resonators for THz and infrared detection. This investigation is an important step toward improving uncooled detectors for terahertz/infrared technologies.
In general, the experimental data is well presented. Overall the manuscript should be clear for the reader. However, in my opinion, the authors should address the following comments before the paper is accepted for publication in the MDPI Sensors journal:
-Authors should more clearly describe novelties in the presented results in comparison to work that was presented in 2019 : Zhang, Ya, et al. "Enhanced thermal sensitivity of MEMS bolometers integrated with nanometer-scale hole array structures." AIP Advances 9.8 (2019): 085102 || doi.org/10.1063/1.5113521
Some estimations of NEP improvement could be interesting for readers.
-The authors present a review of existing solutions and competitive technology, but they also should give more information about the general principles of the presented MEMS beam resonator-detector. This is quite a weak point of the article. One paragraph in the introduction could prepare the reader to evaluate results and familiarize with the basic principle of converting absorbed power into the signal at a lock-in amplifier. It will do more understandable material in the following sections
-Please include additional information about utilized software for tension and thermal properties modeling and analysis.
-in lines 217 and 223: Index D in τD should be corrected to subscript also, τD introduction should be replaced from line 223 to line 217
Round 2
Reviewer 2 Report
Dear authors and editor,
I apologize for the delay with my answer.
I would like to thank the authors for addressing the comments in my initial review. The added manuscript notes are important to a clear understanding of the presented results. I recommend the manuscript for publication.